# Potency of commonly retailed antibiotics in pharmacies found in Adama, Oromia regional state, Ethiopia

Demelash Demissie[1], Teshome Geremew[1], Adinew Zewdu Chernet[2], Musa Mohammed Ali[3]*

1 Adama Science and Technology University, Adama, Ethiopia, 2 Ethiopian Public Health Institute, Addis Ababa, Ethiopia, 3 College of Medicine and Health Sciences, School of Medical Laboratory Science, Hawassa University, Awasa, Ethiopia

* ysnmss@yahoo.com

**Data Availability Statement:** All data and materials are available with in the manuscript.

**Funding:** The author(s) received no specific funding for this work.

## Abstract

### Introduction

Antibiotics are commonly used for the treatment and prevention of bacterial infections. The potency of antibiotics can be affected by factors such as temperature, light, moisture, and storage conditions. Inappropriate storage and transportation of antibiotics may lead to loss of potency earlier than the expiry date. The aim of this study was to determine the potency and associated factors of commonly retailed antibiotics.

### Method

Institution-based cross-sectional study was conducted on commonly retailed antibiotics in pharmacies that are available in Adama, Ethiopia from March 2018 to June 2018. This study focused on commonly ordered antibiotics such as amoxicillin, azithromycin, ciprofloxacin, and ceftriaxone. Antibiotics to be tested were selected by using a simple random sampling technique. Socio-demographic and related data were collected using a semi-structured questionnaire. Antibiotic susceptibility testing was performed using the disc diffusion method as described in the Clinical Laboratory Standard Institute guideline.

### Results

Mean inhibition zones of amoxicillin, ciprofloxacin, azithromycin, and ceftriaxone were $14.2 \pm 4$ mm, $30.9 \pm 4.2$ mm, $17.47 \pm 3.83$ mm, and $32.7 \pm 1.8$ respectively. Out of 164 antibiotics tested, 61% passed the potency test. The potency of antibiotics varies across different countries in which 53.7% and 54.6 of antibiotics from India and Ethiopia passed the potency test. All ceftriaxone tested in this study passed the potency test. Factors such as air condition of pharmacy ($X^2 = 4.27$; $p = 0.039$), source of all antibiotics ($X^2 = 5.41$; $p = 0.02$), and source of amoxicillin ($X^2 = 4.73$; $p = 0.03$) were significantly associated with potency of antibiotics.

**Competing interests:** No authors have competing interests.

**Abbreviations:** CLSI, Clinical Laboratory Standard Institute; OPHL, Oromia Public Health Laboratory; ATCC, American type culture collection; SPSS, Statistical Package for Social Sciences.

## Conclusions

About 40% of antibiotics tested in the current study did not pass the potency test; this warrants further investigation to identify the magnitude of the problem and its causes at a large scale.

## Introduction

Antibiotics have been used for the treatment and prevention of bacterial infections [1]. The effectiveness of antibiotics can be affected by temperature, light, moisture, and storage condition [2]. Counterfeit and substandard antibiotics can also affect the quality of antibiotics. Counterfeit antibiotics are deliberately mislabeled with a fake identity, composition, and source. This can affect generic and branded antibiotics and may include wrong ingredients, no active pharmaceutical ingredients (APIs), and insufficient ingredients. And it also includes antibiotics which are sold with fake labeling and fake packaging [3].

Substandard antibiotics are produced by manufacturers authorized by the national medicine regulatory authority which does not meet the specifications set by national standards [4]. A mild difference in the concentration of the active ingredient during antibiotic preparation may cause allergic reactions, unexpected side effects, poisoning, untreated disease, and early death [5,6]. Careful *in vitro* evaluations are necessary at regular intervals to check the potency of antibiotics sold in the pharmacies [7].

Despite the role of antibiotics in treating the disease; ineffective antibiotics can pose great risks to individuals and even threaten the lives of consumers [8]. Degradation of antibiotics occurs much before they approach the expiry dates because of several factors such as exposure to light and moisture [9]. The loss of potency during storage may influence the efficacy and safety of antibiotics [10].

Poor-quality antibiotics can reach the market through substandard production of legitimate antibiotics due to inadequate quality-control processes during manufacturing, as well as by deliberate fraudulent practices [11]. The wide use of antibiotics with poor quality can lead to the emergence and spread of antibiotic-resistant strains of bacteria in the community [12]. Infection caused by antibiotic-resistant bacteria are more difficult to treat, can lead to long-term illness, increase healthcare costs, and death [9].

Expansion of pharmaceutical industries in many countries with advancement in transportation technologies facilitated not only trade of genuine pharmaceutical products but also the circulation of poor-quality and counterfeited antibiotics in list-income countries [13]. Adama is situated along the road that connects Addis Ababa with Dire-Dawa city. A large number of trucks use the same route to travel to and from the seaports of Djibouti and additionally, the new Addis Ababa-Djibouti railway runs through Adama. This makes Adam one of the busiest cities in the country with strong business activities and several kinds of illegal commodities including counterfeit antibiotics that might be sold in pharmacies found in Adama city. The aim of this study was to determine the potency and associated factors of commonly sold antibiotics in pharmacies that are found in Adama city.

## Materials and methods

### Study setting

This study was conducted in pharmacies located in Adama, Oromia Regional State, Ethiopia. Adama is situated at 99 km to the southeast of Addis Ababa at an altitude of 1712 meters above sea level between the base of an escarpment to the west, and the Great Rift Valley to the East.

### Study design and study population

An institution-based cross-sectional study was conducted on commonly retailed antibiotics from March 2018 to June 2018. In this study, all pharmacies in Adama were included. A total of 164 antibiotics belonging to four categories of antibiotics were included. These antibiotics were amoxicillin (n = 42), ciprofloxacin (n = 42), azithromycin (n = 38), and ceftriaxone (n = 42). From each pharmacy, one antibiotic was randomly selected. Four pharmacies did not have azithromycin. Antibiotics were selected because they are commonly ordered in the study area. Amoxicillin was in the form of a capsule (one stripe contains 10 capsules); ciprofloxacin was in the form of tablets (one stripe contains 10 tablets), and azithromycin was in the form of tablet (one stripe contains 3 tablets). Ceftriaxone was in the form of vials.

### Recommended storage condition of evaluated antibiotics

During the study period, all antibiotics tested in this study were stored in a room with good ventilation and a temperature range of 2–30˚C.

### Study variables

Dependent variable: Potency of antibiotics.

Independent variables: Type of pharmacy, air condition of pharmacy, the shelf life of antibiotics, and source of antibiotics.

### Data collection

We developed a semi-structured questionnaire to collect socio-demographic characteristics of dispensers and the condition of antibiotics and pharmacies. From each pharmacy, one pharmacist or druggist was interviewed. All selected antibiotics were checked for batch numbers, manufacturer, and expiry date. After the assessment, antibiotics were transported to the Oromia Public Health Laboratory (OPHL) for antimicrobial susceptibility testing.

### Antibiotic disc preparation

To prepare antibiotic disc we followed the procedure suggested by Clinical Laboratory Standard Institute (CLSI) [14]. Discs with 6 mm diameter were prepared by punching a sheet of Whatman Number 3 filter paper (United Kingdom) using a perforator. To obtain 10 μg of amoxicillin disc, 500 mg of amoxicillin tablet was dissolved in 250 ml of phosphate buffer and 5 μL of dissolved stock antibiotic was impregnated onto the 6 mm sized disc. To obtain 5 μg of ciprofloxacin disc, 500 mg of ciprofloxacin tablet was dissolved in 500 ml of distilled water and 5 μL of dissolved stock antibiotic was impregnated onto the 6 mm sized disc. To obtain 30 μg of ceftriaxone, 1000 mg of injectable ceftriaxone was dissolved in 166.7 ml of distilled water and 5 μL of dissolved stock antibiotic was impregnated onto the 6 mm sized disc. To obtain 15 μg of azithromycin, 500 mg of azithromycin tablet was dissolved in a 166.7 ml of 95% ethanol with a broth media and 5 μL of dissolved stock antibiotic was impregnated onto the 6 mm sized disc (S1 Table in S1 File). To assess the potency of antibiotics, two reference strains such as *Escherichia coli* (ATCC-25922) and *Staphylococcus aureus* (ATCC-25923) were used.

### Antimicrobial susceptibility testing

The antimicrobial susceptibility testing was performed following the Kirby-Bauer disc diffusion method as described in the CLSI [14]. Briefly, the reference strains (*E. coli* and *S. aureus*) were sub-cultured onto blood agar and incubated at 37˚C overnight. Using a sterile wire loop, 3–5 pure colonies were emulsified in 5 mL of normal saline until the turbidity matches 0.5

McFarland standard. Using a sterile dry cotton swab, bacterial suspensions were uniformly inoculated onto the entire surface of Muller Hinton agar (MHA). Antibiotics disks were placed on the surface of MHA and incubated aerobically at 37˚C for 16–18 hours. The diameter of the zone of inhibition was measured using a ruler; isolates were classified as potent (pass) or not potent (fail) based on the CLSI cut point [14]. All experiments were conducted in triplicates and the means were recorded to determine the potency of antibiotics.

### Pass criteria based on the susceptibility range set by CLSI

Amoxicillin: It is considered as 'pass' if the diameter of the zone of inhibition falls within or greater than the susceptibility range of 15–22 mm.

Ciprofloxacin: It is considered as 'pass' if the diameter of the zone of inhibition falls within or greater than the susceptibility range of 30–40 mm.

Azithromycin: It is considered as 'pass' if the diameter of the zone of inhibition falls within or greater than the susceptibility range of 21–26 mm.

Ceftriaxone: It is considered as 'pass' if the diameter of the zone of inhibition falls within or greater than the susceptibility range of 29–35 mm.

### Data quality control

The semi-questionnaire was validated and pretested before use in the actual study. All laboratory tests were performed according to the CLSI guideline. The sterility and performance of the culture media were checked; the performances of all reagents and materials were checked and ensured. Prior to use, all discs were sterilized and the sterility was checked by placing 8 randomly selected discs on uninoculated MHA agar and incubated at 37˚C for 18 hours. Selected discs impregnated with only solvent were checked for antibacterial activity. The disc diffusion method used in the current study was verified by running the method 20 times using standard antibiotics obtained from OPH: Amoxicillin (10 μg), ciprofloxacin (5 μg), azithromycin (15 μg), and ceftriaxone (30 μg). The precisions for amoxicillin, ciprofloxacin, azithromycin, and ceftriaxone were 2.364 mm, 3.364 mm, 1.63 mm, and 1.67 mm respectively. The uncertainty measurement for amoxicillin, ciprofloxacin, azithromycin, and ceftriaxone with a 95% CI were 17–19 mm, 31–32 mm, 21–24 mm, and 31–33 mm respectively.

### Data management and analysis

Data were checked, coded, and entered into the computer using EP INFO version 7 and analyzed by using a Statistical Package for Social Sciences (SPSS) version 20. Data were analyzed using Chi-square ($X^2$); a $p$-value of $< 0.05$ was considered as statistically significant.

## Results

### Socio-demographic data

In this study, a total of 42 pharmacies were included. Out of which, 81% belongs to the private and 19% belongs to the government. A total of 42 dispensers (13 males and 29 females) were interviewed. The ages of 45.2% of dispensers were between 31–40 years. The work experience of 54.8% of dispensers was 5–9 years (Table 1).

**Characteristics and source of antibiotics.** Most amoxicillin was manufactured in Ethiopia (73.8%) whereas most ciprofloxacin was manufactured in South Korea and Romania. Most of the ceftriaxone were manufactured in China (Table 2).

**Table 1. Characteristics of dispensers and types of Pharmacy at Adama, Oromia, Ethiopia, March 2018 to June 2018 (N = 42).**

| Variables | Category | n (%) |
|---|---|---|
| Type of Pharmacy | Private | 34 (81) |
| | Government | 8 (19) |
| Sex of dispensers | Male | 13 (31) |
| | Female | 29 (69) |
| Age of dispensers in years | ≤25 | 4 (9.5) |
| | 26–30 | 15 (35.7) |
| | 31–40 | 19 (45.2) |
| | 41–49 | 4 (9.5) |
| | ≥50 | - |
| Qualification of dispensers | Druggist | 8 (19) |
| | Pharmacist | 34 (81) |
| Work experience of dispensers in years | <1 | - |
| | 1–4 | 6 (14.3) |
| | 5–9 | 23 (54.8) |
| | 10–14 | 9 (21.4) |
| | ≥15 | 4 (9.5) |

## Shelf life and handling conditions of antibiotics

Almost all antibiotics retailed in the private and government pharmacies in Adama had a long expiry date (> six months). At the time of collection, none of the antibiotics were expired. All amoxicillin and ciprofloxacin had a long expiry date. Thirty (78.9%) and 38 (90.5) azithromycin and ceftriaxone had a long expiry date respectively. Forty (90.5%) of the pharmacies had enough storage area. Out of 42 pharmacies, 25 (59.5%) had ventilator with good working conditions, 32 (76.2%) had refrigerator and 26 (61.9%), and 25 (59.5%) of them had a thermometer in dispenser and store room respectively. Thirty-eight (90.5%) of the respondents replied that there was an inspection by a regulatory body. Thirty-one (81.6%) of respondents replied that inspection was made in three months intervals and 7 (18.4%) replied that inspection was made in six months intervals.

## Mean inhibition zones of antibiotics

Mean inhibition zones of amoxicillin, ciprofloxacin, azithromycin, and ceftriaxone were 14.2 ± 4 mm, 30.9 ± 4.2 mm, 17.47 ± 3.83 mm, and 32.7±1.8 respectively. Most of the amoxicillin from Ethiopia (EPHARM-Ethiopia) had a mean zone of inhibition of less than 12 mm (Table 3).

## Potency of antibiotics

Out of 164 antibiotics tested in this study, 61% were potent. 53.7% and 54.6 of antibiotics from India and Ethiopia were potent respectively (Table 4). The total potency of amoxicillin, ciprofloxacin, azithromycin, and ceftriaxone were 50%, 66.7%, 23.7%, and 100% respectively (Table 5).

## Factors that influence the potency of antibiotics

In the current study, among different factors assessed air condition of the pharmacy for amoxicillin ($X^2$ = 4.27; $p$ = 0.039), the source of all antibiotics ($X^2$ = 5.41; $p$ = 0.02), and the source of

**Table 2. Frequency of antibiotics based on manufacturers at Adama, Ethiopia, March 2018 to June 2018.**

| Antibiotics | Manufacturer | n (%) |
|---|---|---|
| Amoxicillin | APF (Ethiopia) | 15 (35.7) |
| | DENK-Pharma (Germany) | 1 (2.4) |
| | EPHARM (Ethiopia) | 16 (38.1) |
| | GlaxoSmithKline (UK) | 3 (7.1) |
| | Kopra Ltd (India) | 6 (14.3) |
| | Remedica (Cyprus) | 1 (2.4) |
| | Total | 42 (100) |
| Ciprofloxacin | APF (Ethiopia) | 4 (9.5) |
| | Brawn Lab (India) | 1 (2.4) |
| | Cadila Pharmaceutical (Ethiopia) | 4 (9.5) |
| | EPHARM(Ethiopia) | 6 (14.3) |
| | Huonsco Ltd (S. Korea) | 10 (23.8) |
| | Leben laboratories (India) | 7 (16.7) |
| | Sandoz a Navartis Company (Romania) | 10 (23.8) |
| | Total | 42 (100) |
| Azithromycin | Bafna Pharmaceutical Ltd (India) | 1 (2.6) |
| | Beximco pharmaceutical Ltd (Bangladesh) | 7 (18.4) |
| | Cadila Pharmaceuticals (Ethiopia) | 9 (23.7) |
| | Coral Laboratories Ltd (India) | 2 (5.3) |
| | Corporation Ltd (Kenya) | 1 (2.6) |
| | Deva Holding (Turkey) | 5 (13.2) |
| | EPHARM (Ethiopia) | 1 (2.6) |
| | Sandoz a Navartis company (Romania) | 2 (5.3) |
| | Sherya life science (India) | 1 (2.6) |
| | Umedica Laboratories Ltd (India) | 1 (2.6) |
| | ZIM Laboratories Ltd (India) | 8 (21.1) |
| | Total | 38 (100) |
| Ceftriaxone | Ashish Life Science (India) | 6 (14.3) |
| | AsralSteriTech PLC Ltd (India) | 5 (11.9) |
| | BilimPharmaceutical (Turkey) | 2 (4.8) |
| | CSPC ZhougnuoPharm (China) | 9 (21.4) |
| | Gulf Pharmaceuticalind (UAE) | 3 (7.1) |
| | Shandonglukang pharmaceutical (China) | 13 (30.9) |
| | Theon Pharmaceutical (India) | 2 (4.8) |
| | VHB medi science ltd (India) | 1 (2.4) |
| | Zhuhai Kinhoo pharmaceutical (China) | 1 (2.4) |
| Total | | 42 (100) |

EPHARM: Ethiopian pharmaceutical manufacturing, APF: Addis pharmaceutical factory plc (private limited company), UAE: United Arab Emirates, UK: United Kingdom.

amoxicillin ($X^2 = 4.73$; $p = 0.03$) were significantly associated with potency of antibiotics (Table 6).

## Discussion

Out of 164 antibiotics tested in this study, only 61% were found to be potent. The potency of antibiotics varies based on the countries of origin in which 53.7% and 54.6% of antibiotics from India and Ethiopia were potent respectively. Among the four categories of antibiotics

**Table 3. Mean zone of inhibition of selected antibiotics collected from pharmacies found in Adama, Ethiopia, March 2018 to June 2018.**

| Bacteria | Amoxicillin manufacture | n | Mean ZI (mm±SD) | Standard range(mm) |
|---|---|---|---|---|
| E. coli (E. coli ATCC 25922) | APF-Ethiopia | 15 | 14.73±3.4 | 15–22 |
| | DENK Pharma-Germany | 1 | 15 ±0 | |
| | EPHARM-Ethiopia | 16 | 11.88±4.7 | |
| | GSK-London | 3 | 14.67± 3.2 | |
| | KOPRA LIMTED-India | 6 | 21.33±2.3 | |
| | Remedica-Cyprus | 1 | 15±0 | |
| | Mean IZ of Amoxicillin | 42 | 14.2±4.0 | |
| | **Ciprofloxacin manufacturer and country** | | | |
| E. coli (ATCC 25922) | APF-Ethiopia | 4 | 34±7.1 | 30–40 |
| | BRAWN Lab.-India | 1 | 32±0 | |
| | CADILA Pharmaceutical- Ethiopia | 4 | 34±2.8 | |
| | EPHARM-Ethiopia | 6 | 32.5±4.6 | |
| | Huonsco Ltd- S. korea | 10 | 31.1±1.9 | |
| | Leben lab.-India | 7 | 26.3±8.2 | |
| | SANDOZ aNavartis-Romania | 10 | 29.1±3.5 | |
| | Mean IZ of ciprofloxacin | 42 | 30.9±4.2 | |
| | **Azithromycin manufacture and country** | | | |
| S. aureus (ATCC 25923) | Bafna Pharmaceutical-India | 1 | 12±0 | 21–26 |
| | Beximico pharmaceutical -Bangladesh | 7 | 16.5±4.57 | |
| | CADILA Pharmaceutical -Ethiopia | 9 | 18.5±3.5 | |
| | Coral Lab.-India | 2 | 19.33±4.18 | |
| | Corporation limited-Kenya | 1 | 20±0 | |
| | Deva Holding-Turkey | 5 | 19.33±4.18 | |
| | EPHARM-Ethiopia | 1 | 14±0 | |
| | Sandoz–Romania | 2 | 17±4.2 | |
| | SHERYA LIFE Science- India | 1 | 20±0 | |
| | Umedica Laboratory-India | 1 | 14±0 | |
| | ZIM lab.-India | 8 | 16.12±3.5 | |
| | Mean ZI of Azithromycin | 38 | 17.47±3.83 | |
| | **Ceftriaxone manufacturer and country** | | | |
| E. coli (ATCC 25922) | Ashish life science-India | 6 | 33±1.7 | 29–35 |
| | Asralsteril Tech-India | 5 | 32.4±1.3 | |
| | BILIM pharmaceutical- Turkey | 2 | 34±1.4 | |
| | CSPC Zhougnuo Pharmaceutical-China | 9 | 33.2±1.4 | |
| | Gulf pharmaceutical industry-UAE | 3 | 32±1.7 | |
| | Shandonglukang pharmaceutical-China | 13 | 32.5±3 | |
| | Theon Pharmaceutical-India | 2 | 32±1.4 | |
| | VHB MEDI Science limited-India | 1 | 35±0 | |
| | Zhuhai Kinhoo pharmaceutical- China | 1 | 33±0 | |
| | Mean IZ of ceftriaxone | 42 | 32.7±1.8 | |

Mm: millimeter, SD: Standard deviation, ZI: Zone of inhibition, ATCC: American type culture collection, UAE: United Arab Emirates.

tested, less than 50% of azithromycin were not potent with a mean inhibition zone diameter of 17.47±3.83 mm. The inhibition zone was unsatisfactory according to the standard inhibition zone set by the CLSI guideline (21–26 mm) [15]. The inhibition zone of azithromycin in this study was probably affected by the pH of the microbiological growth medium and its dissolubility in 95% ethanol alcohol diluents [16].

**Table 4. Potency of selected antibiotics collected from pharmacies found in Adama, Ethiopia based on the source country, March 2018 to June 2018.**

| Country | Pass n (%) | Fail, n (%) | Total |
|---|---|---|---|
| India | 22 (53.7) | 19 (46.3) | 41 |
| Bangladesh | - | 7 (100) | 7 |
| Ethiopia | 30 (54.6) | 25 (45.5) | 55 |
| Kenya | - | 1 (100) | 1 |
| Turkey | 6 (85.7) | 1(14.3) | 7 |
| Romania | 3 (25) | 9 (75) | 12 |
| Germany | 1 (100) | - | 1 |
| United Kingdom | 2 (66.7) | 1 (33.3) | 3 |
| Cyprus | 1 (100) | - | 1 |
| S. Korea | 9 (90) | 1 (10) | 10 |
| China | 23 (100) | - | 23 |
| UAE | 3 (100) | - | 3 |
| **Total** | **100 (61)** | **64 (39)** | **164** |

UAE: United Arab Emirates, S. Korea: South Korea.

About 50% of amoxicillin and ciprofloxacin in the current study were not potent indicating some defect either in the antibiotics itself, storage, limitation of our laboratory methods, and antibiotics can be counterfeit or of substandard [17]. Similarly, bacteria isolated from the clinical specimen was reported to have reduced susceptibility to amoxicillin and ciprofloxacin [18,19]. Unlike this study, most of the amoxicillin tested in the Middle East was not counterfeit and contained the amount of ingredients claimed by the manufacturer [20,21].

This study revealed that the mean inhibition zone diameter of amoxicillin to be 14.2 ± 4 mm which is not in line with the finding reported from Ghana (16 ± 0.6mm) [22]. Moreover, most of the amoxicillin produced in Ethiopia (EPHARM) had a mean zone of inhibition of less than 12 mm which is below the standard set by the CLSI. The variation observed could be due to a difference in the manufacturer; most of the amoxicillin in this study was locally produced. The difference could also be attributed to the laboratory method used to determine the potency of antibiotics.

The mean inhibition zone of ciprofloxacin was 30.9 ± 4.2mm which is in the range set by CLSI. However, most of them were not potent. In this study, the mean inhibition zone of ceftriaxone was 32.7±1. According to CLSI inhibition zone range, all ceftriaxone antibiotics were potent. Several studies, which were conducted on different clinical isolates, indicated the effectiveness of ceftriaxone [18,19,23].

The potency of antibiotics can be affected by various factors. In the present study, the air condition of the pharmacy ($X^2 = 4.27$; $p = 0.039$) and source of the antibiotics were significantly associated with the potency of antibiotics ($X^2 = 5.41$; $p = 0.02$). According to this study, 52% of all antibiotics from private pharmacies were not potent. About 64% of amoxicillin stored in a pharmacy with no ventilator failed to pass the potency test indicating the importance of the ventilator in keeping the integrity of antibiotics stored in the pharmacy. Most of the antibiotics (46.3%) from the local source (Ethiopia) were not potent; specifically, 54.8% of amoxicillin from the local source was unable to pass the potency test ($X^2 = 4.73$; $p = 0.03$). This could be due to problems that could arise during the production, shipment, and storage condition of antibiotics. Moreover, antibiotics could be of substandard or counterfeit which may contain active ingredients below the required level.

**Table 5. Potency of selected antibiotics collected from pharmacies found in Adama, Ethiopia based on the manufacturer and country, March 2018 to June 2018.**

| Amoxicillin | | | | | | |
|---|---|---|---|---|---|---|
| **Manufacturer** | **Country** | **Brand name** | **Concentration** | **Pass, n (%)** | **Fail, n (%)** | **Total** |
| APF | Ethiopia | Amoxid | 500mg | 6(40) | 9 (60) | 15 |
| DENK Pharma | Germany | Amoxi-Denk | 500mg | 1 (100) | - | 1 |
| EPHARM | Ethiopia | Amoxicillin | 500mg | 8 (50) | 8 (50) | 16 |
| GlaxoSmithKline | UK | Amoxil | 500mg | 2 (66.7) | 1 (33.3) | 3 |
| KOPRA limited | India | AMYN | 500mg | 3 (50) | 3(50) | 6 |
| Remedica | Cyprus | Amoxapen | 500mg | 1 (100) | - | 1 |
| Total | | | | 21 (50) | 21 (50) | 42 |
| **Ciprofloxacin** | | | | | | |
| APF | Ethiopia | CIF LOX | 500mg | 3 (75) | 1 (25) | 4 |
| BRAWN Laboratory | India | Brucipro | 500mg | 1 (100) | - | 1 |
| CADILA Pharmaceutical | Ethiopia | Ciprodac | 500mg | 4 (100) | - | 4 |
| EPHARM | Ethiopia | Ciprofloxacin | 500mg | 5 (88.3) | 1 (16.7) | 6 |
| Huonsco Ltd | S. Korea | Floxine | 500mg | 9 (90) | 1 (10) | 10 |
| Leben laboratory | India | Ciproleb | 500mg | 3 (42.9) | 4 (57.1) | 7 |
| SANDOZ aNavartis comp | Romania | Serviflox | 500mg | 3 (30) | 7 (70) | 10 |
| Total | | | | 28 (66.7) | 14 (33,3) | 42 |
| **Azithromycin** | | | | | | |
| Bafna Pharmaceutical | India | AZIBIAL | 500mg | - | 1 (100) | 1 |
| Beximico pharmaceutical | Bangladesh | Azithrocin | 500mg | - | 7(100%) | 7 |
| CADILA Pharmaceutical | Ethiopia | Zycin | 500mg | 4 (44.4) | 5 (55.6) | 9 |
| Coral Laboratory | India | Cortzite | 500mg | - | 2 (100) | 2 |
| Corporation limited | Kenya | THROZA | 500mg | - | 1 (100) | 1 |
| Deva Holding | Turkey | Azitro | 500mg | 4 ((80) | 1 (20) | 5 |
| EPHARM | Ethiopia | Ephazit | 500mg | - | 1 (100) | 1 |
| Sandoz | Romania | Binozyt | 500mg | - | 2 (100) | 2 |
| SHERYA LIFE Science | India | ZIT | 500mg | - | 1 (100) | 1 |
| Umedica Laboratory | India | UZET | 500mg | - | 1 (100) | 1 |
| ZIM Laboratory | India | Azito | 500mg | 1 (12.5) | 7 (87.5) | 8 |
| Total | | | | 9 (23.7) | 29 (76.3) | 38 |
| **Ceftriaxone** | | | | | | |
| Ashish life science | India | CEFTASH | 1000mg | 6 (100) | - | 6 |
| Asralsteril Tech | India | ZEFONE100 | 1000mg | 5 (100 | - | 5 |
| BILIM pharmaceutical | Turkey | FORSEF | 1000mg | 2 (100) | - | 2 |
| CSPC Zhougnuo Pharmaceutical | China | Ceftazone | 1000mg | 9 (100) | - | 9 |
| Gulf pharmaceutical industry | UAE | Triaxon | 1000mg | 3 (100) | - | 3 |
| Shandonglukang pharmaceutical | China | Ceftriaxone | 1000mg | 13 (100) | - | 13 |
| Theon Pharmaceutical | India | THEOXONE | 1000mg | 2 (100) | - | 2 |
| VHB MEDI Science limited | India | IVIXONE | 1000mg | 1 (100) | - | 1 |
| Zhuhai Kinhoo pharmaceutical | China | Ceftriaxone | 1000mg | 1 (100) | - | 1 |
| Total | | | | 42 (100) | - | 42 |

UAE: United Arab Emirates, UK: United Kingdom, APF: Addis pharmaceutical factory.

## Limitation of the study

Only four antibiotics were tested, which might not be representative of other antibiotics. We faced a shortage of similar studies to compare the findings of this study.

**Table 6. Factors that affect the potency of selected antibiotics collected from pharmacies found in Adama, Ethiopia (March 2018 to June 2018).**

| Variables | | Potency of antibiotics* | | $X^2$ value | p-value |
|---|---|---|---|---|---|
| | | Pass, n (%) | Fail, n (%) | | |
| **Type of pharmacy from where all antibiotics obtained** | **Private** | **49 (48)** | **53 (52)** | **3.47** | **0.07** |
| | **Government** | **14 (70)** | **6 (30)** | | |
| Type of pharmacy from where amoxicillin was obtained | Private | 18 (52.9) | 16 (47.1) | 0.24 | 0.63 |
| | Government | 5 (62.5) | 3 (37.5) | | |
| Type of pharmacy from where ciprofloxacin was obtained | Private | 20 (58.8) | 14 (41.2) | 3.68 | 0.06 |
| | Government | 8 (100) | - | | |
| Type of pharmacy from where azithromycin was obtained | Private | 11 (32.4) | 23 (67.6) | 0.09 | 0.77 |
| | Government | 1 (25) | 3 (75) | | |
| **Air condition of pharmacy for all antibiotics** | **Ventilation** | **40 (55.6)** | **32 (44.4)** | **1.07** | **0.30** |
| | **No ventilation** | **23 (46)** | **27 (54)** | | |
| Air condition of pharmacy for amoxicillin | Ventilation | 17 (68) | 8 (32) | 4.27 | 0.039 |
| | No ventilation | 6 (35.3) | 11 (64.7) | | |
| Air condition of pharmacy for ciprofloxacin | Ventilation | 16 (64) | 9 (36) | 0.19 | 0.66 |
| | No ventilation | 12 (70.6) | 5 (29.4) | | |
| Air condition of pharmacy for azithromycin | Ventilation | 7 (31.8) | 15 (68.2) | 0.001 | 0.97 |
| | No ventilation | 5 (31.3) | 11 (68.7) | | |
| **Shelf life for all antibiotics** | Long expired date | 84 (55.3) | 68 (44.7) | 0.16 | 0.69 |
| | Near to expiry (<6mnth) | 4 (33.3) | 8 (66.7) | | |
| Shelf life for azithromycin | Long Expiry date | 9 (30) | 21 (70) | 0.160 | 0.69 |
| | Near to expiry date (<6mnth) | 3 (37.5) | 5 (62.5) | | |
| **Source for all antibiotics** | Local | 29 (53.7) | 25 (46.3) | 5.41 | **0.02** |
| | Foreign | 76 69.1) | 34 (30.9) | | |
| Source for amoxicillin | Local | 14 (45.2) | 17 (54.8) | 4.73 | 0.03 |
| | Foreign | 9 (81.8) | 2 (18.2) | | |
| Source for ciprofloxacin | Local | 11 (84.6) | 2 (15.4) | 2.97 | 0.85 |
| | Foreign | 17 (58.6) | 12 (41.4) | | |
| Source for azithromycin | Local | 4 (40) | 6 (60) | 1.67 | 0.51 |
| | Foreign | 8 (28.6) | 20 (71.4) | | |
| Awareness on unlicensed antibiotic distributer | Yes | 42 (53.2) | 37 (46.8) | 0.001 | 0.97 |
| | No | 46 (54.1) | 39 (45.9%) | | |
| Work experience | < 10years | 64 (52.9) | 57 (47.1) | 0.023 | 0.88 |
| | ≥10 years | 24(55.8) | 19 (44.2%) | | |

* Ceftriaxone was excluded from analysis because all of them (n = 42) were potent.

## Conclusions

About 60% of antibiotics tested in the current passed the potency test. The potency of antibiotics varies based on the source country where 53.7% and 54.6 of antibiotics from India and Ethiopia passed the potency test. Most of the azithromycin did not pass the potency test while all ceftriaxone passed the potency test. Among factors assessed, air condition of pharmacy, source of all antibiotics, and source of amoxicillin were significantly associated with potency of antibiotics. This study highlights the importance of further investigation to identify the magnitude of the problem and its causes at a large scale.

## Supporting information

**S1 File.**
(DOCX)

## Acknowledgments

We would like to acknowledge the staff of the Oromia public health laboratory for facilitating laboratory work. We also acknowledge study participants for their willingness to participate in the study.

## Author Contributions

**Conceptualization:** Demelash Demissie, Teshome Geremew, Adinew Zewdu Chernet.

**Data curation:** Demelash Demissie.

**Formal analysis:** Demelash Demissie, Musa Mohammed Ali.

**Investigation:** Demelash Demissie, Adinew Zewdu Chernet.

**Methodology:** Demelash Demissie, Teshome Geremew, Adinew Zewdu Chernet, Musa Mohammed Ali.

**Supervision:** Adinew Zewdu Chernet, Musa Mohammed Ali.

**Validation:** Teshome Geremew.

**Visualization:** Teshome Geremew.

**Writing – original draft:** Demelash Demissie, Adinew Zewdu Chernet, Musa Mohammed Ali.

**Writing – review & editing:** Musa Mohammed Ali.

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
