## [Decision Letter · Decision Letter 0]

23 Apr 2021

PONE-D-21-05088

Potency of commonly retailed antibiotics in pharmacies found in Adama, Oromia regional state, Ethiopia

PLOS ONE

Dear Dr. Ali,

Thank you for submitting your manuscript to PLOS ONE. After careful consideration, we feel that it has merit but does not fully meet PLOS ONE’s publication criteria as it currently stands. Therefore, we invite you to submit a revised version of the manuscript that addresses the points raised during the review process.

We look forward to receiving your revised manuscript.

Kind regards,

Praveen Rishi, Ph.D., FAMI, FABMS

Academic Editor

PLOS ONE

Journal Requirements:

Please include additional information regarding the survey or questionnaire used in the study and ensure that you have provided sufficient details that others could replicate the analyses. For instance, if you developed a questionnaire as part of this study and it is not under a copyright more restrictive than CC-BY, please include a copy, in both the original language and English, as Supporting Information. Moreover, please include more details on how the questionnaire was pre-tested, and whether it was validated. Moreover, please clarify how each variable was defined and categorised.

Reviewers' comments:

Reviewer's Responses to Questions

**Comments to the Author**

1. Is the manuscript technically sound, and do the data support the conclusions?

Reviewer #1: No

Reviewer #2: Partly

2. Has the statistical analysis been performed appropriately and rigorously? 

Reviewer #1: No

Reviewer #2: I Don't Know

3. Have the authors made all data underlying the findings in their manuscript fully available?

Reviewer #1: Yes

Reviewer #2: Yes

4. Is the manuscript presented in an intelligible fashion and written in standard English?

Reviewer #1: No

Reviewer #2: No

5. Review Comments to the Author

Reviewer #1: In this submitted manuscript   authors  claimed that about 40% of antibiotics tested in the present study were not potent and the potency of antibiotics varies based on the source of the country.  Authors also claimed that most of azithromycin were not potent while all ceftriaxone were potent.  Author (s) mentioned  that the physicochemical condition in the  pharmacy setup significantly associated with the potency of antibiotics.Manuscript is not technically sound due to poor presentation and   so it is very difficult  to understand what kind of message to give the author (s) in the present MS.   However, there are following  points that need to be addressed :1. What could be the scientific reason,  reduction in potency of antibiotic.?2. Does variation of API ( Source of API) in the antibiotics preparations responsible for antibiotic potency.3. Does the pharmacy set up awareness of this  type of loss in the potency of antibiotics?

Reviewer #2: Comments

This study about “Potency of commonly retailed antibiotics in pharmacies found in Adama, Oromia regional state, Ethiopia -” provides interesting data but has many shortcomings. These points should be addressed. My major comments are as follows.

1.The manuscript is not written so well and therefore, it is difficult to understand what authors want to say.

2.Language of the manuscript needs to be improved thoroughly. Some points, words or sentences have been highlighted in the manuscript and corrections should be made throughout the manuscript thoroughly.

3.On page number 13, E. coli must be italicized in table 3.

4. “To obtain 10 μg of amoxicillin disc, 500 mg of amoxicillin tablet was dissolved in 250 ml of phosphate buffer and 5 μm of dissolved stock antibiotic was impregnated on the 6 mm sized disc. To obtain 5 μg of ciprofloxacin disc, 500mg of ciprofloxacin tablet was dissolved in 500ml of distilled water and 5 μm of dissolved stock antibiotic was impregnated on the 6 mm sized disc. To obtain 30 μg of ceftriaxone, 1000mg of injectable ceftriaxone was dissolved in 166.7ml of distilled water and 5 μm of dissolved stock antibiotic was impregnated on the 6 mm sized disc. To obtain 15 μg of azithromycin, 500mg of azithromycin the tablet was dissolved in a 166.7 ml of 95% ethanol with a broth media and 5 μm of dissolved stock antibiotic was impregnated on the 6 mm sized disc.” Authors have used different volumes of different solvents and different amounts of antibiotics, justify.

5.“5 μm of dissolved stock antibiotic was impregnated on the 6 mm sized disc”. What does “5 μm” mean? How authors confirmed/checked that 5 μm was impregnated on disc?

6.Present work describe the potency of antibiotics. In Table 4, what factors were considered to define/calculate potency of antibiotics on the basis of country?

7.Which factors define the pass and fail percentage of any antibiotic in Table 4 and 5?

6. PLOS authors have the option to publish the peer review history of their article (what does this mean?). If published, this will include your full peer review and any attached files.

Reviewer #1: **Yes: **NA

Reviewer #2: No

---

## [Author Response · Author response to Decision Letter 0]

3 May 2021

Note: HTML markup is below. Please do not edit.]

Reviewers' comments:

Reviewer's Responses to Questions

Comments to the Author

1. Is the manuscript technically sound, and do the data support the conclusions?

Reviewer #1: No

Reviewer #2: Partly

2. Has the statistical analysis been performed appropriately and rigorously?

Reviewer #1: No

Reviewer #2: I Don't Know

3. Have the authors made all data underlying the findings in their manuscript fully available?

Reviewer #1: Yes

Reviewer #2: Yes

4. Is the manuscript presented in an intelligible fashion and written in standard English?

Reviewer #1: No

Reviewer #2: No

5. Review Comments to the Author

Reviewer #1: In this submitted manuscript authors claimed that about 40% of antibiotics tested in the present study were not potent and the potency of antibiotics varies based on the source of the country. Authors also claimed that most of azithromycin were not potent while all ceftriaxone were potent. Author (s) mentioned that the physicochemical condition in the pharmacy setup significantly associated with the potency of antibiotics. Manuscript is not technically sound due to poor presentation and so it is very difficult to understand what kind of message to give the author (s) in the present MS. 

Response: Thank you for your valuable comments. 

• In order to address the technical issue of the manuscript, we have included the following information in the method section: categories of antibiotics tested and their form (in the study design and population), Recommended storage condition of evaluated antibiotics, antibiotic disc preparation heading with detail description, antimicrobial susceptibility testing heading with detail description, pass criteria heading with detail description. We have also included details of what we did to maintain the quality of the data under Data quality control. All these are shown in the manuscript with track change. 

However, there are following points that need to be addressed :1. What could be the scientific reason, reduction in potency of antibiotic.?2. Does variation of API ( Source of API) in the antibiotics preparations responsible for antibiotic potency.3. Does the pharmacy set up awareness of this type of loss in the potency of antibiotics?

Response: 

• Yes, most of the antibiotics tested did not pass the potency test; several factors can be the reason for the failure. In addition to factors that are significantly associated with the potency of the antibiotics in the current study, other factors such as substandard, counterfeit, laboratory methods used can also affect the potency of antibiotics. There are also possibilities of illegal antibiotics in pharmacies. We have mentioned these reasons in the middle and as the end of the discussion section. 

• The pharmacies are of about the ineffectiveness of antibiotics indirectly from customers (there is no improvement despite antibiotic intake). We have communicated the finding of this study to concerned bodies. 

Reviewer #2: Comments

This study about “Potency of commonly retailed antibiotics in pharmacies found in Adama, Oromia regional state, Ethiopia -” provides interesting data but has many shortcomings. These points should be addressed. My major comments are as follows.

1. The manuscript is not written so well and therefore, it is difficult to understand what authors want to say.

Response: Thank you for your constructive comments.

• We agree with comment; we have revised the manuscript as shown in the track change.

2.Language of the manuscript needs to be improved thoroughly. Some points, words or sentences have been highlighted in the manuscript and corrections should be made throughout the manuscript 

Response:

• We agree with comment; we have revised the languages as shown in the track change.

3.On page number 13, E. coli must be italicized in table 3.

Response: We italicized as suggested.

4. “To obtain 10 μg of amoxicillin disc, 500 mg of amoxicillin tablet was dissolved in 250 ml of phosphate buffer and 5 μm of dissolved stock antibiotic was impregnated on the 6 mm sized disc. To obtain 5 μg of ciprofloxacin disc, 500mg of ciprofloxacin tablet was dissolved in 500ml of distilled water and 5 μm of dissolved stock antibiotic was impregnated on the 6 mm sized disc. To obtain 30 μg of ceftriaxone, 1000mg of injectable ceftriaxone was dissolved in 166.7ml of distilled water and 5 μm of dissolved stock antibiotic was impregnated on the 6 mm sized disc. To obtain 15 μg of azithromycin, 500mg of azithromycin the tablet was dissolved in a 166.7 ml of 95% ethanol with a broth media and 5 μm of dissolved stock antibiotic was impregnated on the 6 mm sized disc.” Authors have used different volumes of different solvents and different amounts of antibiotics, justify.

Response: For preparation of antibiotics, we have followed CLSI guideline (CLSI guideline M_100S26, 2016, 26 ed, Table 6A, page 192). Preparation procedure is attached as supplement file. 

5.“5 μm of dissolved stock antibiotic was impregnated on the 6 mm sized disc”. What does “5 μm” mean? How authors confirmed/checked that 5 μm was impregnated on disc?

Response: We agree with the comment and corrected as 5 μL.

6.Present work describe the potency of antibiotics. In Table 4, what factors were considered to define/calculate potency of antibiotics on the basis of country?

Response: We agree with the comment and included ‘pass’ and ‘fail’ criteria set by CLSI in the method section as shown in the track change. 

7.Which factors define the pass and fail percentage of any antibiotic in Table 4 and 5?

Response: We agree with the comment and included ‘pass’ and ‘fail’ criteria set by CLSI in the method section.

---

## [Decision Letter · Decision Letter 1]

17 Jun 2021

Potency of commonly retailed antibiotics in pharmacies found in Adama, Oromia regional state, Ethiopia

PONE-D-21-05088R1

Dear Ali, 

We’re pleased to inform you that your manuscript has been judged scientifically suitable for publication and will be formally accepted for publication once it meets all outstanding technical requirements.

Kind regards,

Praveen Rishi, Ph.D., FAMI, FABMS

Academic Editor

PLOS ONE

Additional Editor Comments (optional):

Reviewers' comments:

Reviewer's Responses to Questions

**Comments to the Author**

1. If the authors have adequately addressed your comments raised in a previous round of review and you feel that this manuscript is now acceptable for publication, you may indicate that here to bypass the “Comments to the Author” section, enter your conflict of interest statement in the “Confidential to Editor” section, and submit your "Accept" recommendation.

Reviewer #1: (No Response)

Reviewer #2: (No Response)

2. Is the manuscript technically sound, and do the data support the conclusions?

Reviewer #1: Partly

Reviewer #2: Partly

3. Has the statistical analysis been performed appropriately and rigorously? 

Reviewer #1: I Don't Know

Reviewer #2: N/A

4. Have the authors made all data underlying the findings in their manuscript fully available?

Reviewer #1: Yes

Reviewer #2: Yes

5. Is the manuscript presented in an intelligible fashion and written in standard English?

Reviewer #1: No

Reviewer #2: Yes

6. Review Comments to the Author

Reviewer #1: Still the author must improve the language of the MS rigorously. Still it is very difficult to understand the MS due to poor presentation.

Reviewer #2: In line 147; “The uncertainty measurement for amoxicillin, ciprofloxacin, azithromycin, and ceftriaxone with a 95% CI were 17-19 mm, 31-32 mm, 21-24 mm, and 31-33 mm respectively.” What does “CI” refer to?

7. PLOS authors have the option to publish the peer review history of their article (what does this mean?). If published, this will include your full peer review and any attached files.

Reviewer #1: **Yes: **Satish K Pandey

Reviewer #2: No

---

## [Editor Report · Acceptance letter]

21 Jun 2021

PONE-D-21-05088R1 

Potency of commonly retailed antibiotics in pharmacies found in Adama, Oromia regional state, Ethiopia 

Dear Dr. Ali:

I'm pleased to inform you that your manuscript has been deemed suitable for publication in PLOS ONE. Congratulations! Your manuscript is now with our production department. 

Kind regards, 

on behalf of

Prof. Praveen Rishi 

Academic Editor

PLOS ONE